# A 3D Discrete-Continuum Coupling Approach for Investigating the Deformation and Failure Mechanism of Tunnels across an Active Fault: A Case Study of Xianglushan Tunnel

**Yalina Ma** [1,2] **, Qian Sheng** [1,2,*] **, Guimin Zhang** [3] **and Zhen Cui** [1,3]

[1] State Key Laboratory of Geomechanics and Geotechnical Engineering, Institute of Rock and Soil Mechanics, Chinese Academy of Sciences, Wuhan 430071, China; mayalina@126.com (Y.M.); zcui@whrsm.ac.cn (Z.C.)

[2] School of Engineering Science, University of Chinese Academy of Sciences, Beijing 100049, China

[3] State Key Laboratory for GeoMechanics and Deep Underground Engineering, China University of Mining & Technology, Xuzhou 221116, China; gmzhang@cumt.edu.cn

\* Correspondence: qsheng@whrsm.ac.cn

**Abstract:** For water transmission tunnels constructed in high-risk seismic regions of western China, active faults pose threats of serious ruptures to the tunnels. To overcome this issue, a 3D discrete-continuum coupling approach is introduced into the study. By this approach, spherical discrete-element-method (DEM) particles are used to represent the surrounding rock mass, and the tunnel is considered to be the continuous finite-difference-method (FDM) zone. In this way, a 3D coupling model was established to study the longitudinal displacement profile and stress response of the tunnel lining under various fault dislocations. The failure pattern of the surrounding rock mass was investigated from a micro perspective. Meanwhile, the design strategy of flexible joint was investigated with the present numerical model. The results from a parametric study show that the smaller segment length, wider width and weaker strength of the flexible joints are beneficial to the anti-dislocation performance of the tunnel. Moreover, an orthogonal array test technique was utilized to investigate the influence level of the main design parameters of the flexible joint on the lining internal stress. With the obtained knowledge, the optimal combination for flexible joint design was presented. Findings may provide references for the anti-dislocation issue of tunnels across active faults.

**Keywords:** tunnel; active fault zone; discrete-continuum coupling method; micromechanics; mechanical response; flexible joint

## 1. Introduction

Active faults, which exist in the weak parts of the earth's crust, are potential sources of seismic activity. When a tunnel inevitably crosses a fault zone, it is directly subjected to strong impacts due to the dislocation of the two parts of an active fault, which leads to the failure of the tunnel lining [1,2]. Previous research has shown that tunnel structures are vulnerable to seismic fault zones with poor quality of the surrounding rock mass and significant changes in the stratigraphic type; examples include the 1906 San Francisco earthquake in the USA [3], the 1995 Hanshin earthquake in Japan [4] and the 1999 Chi-Chi earthquake in Taiwan, China [5–7]. Examples of cross-fault tunnel damage are shown in Table 1. Therefore, an important prerequisite for ensuring the safety of lifeline projects is to investigate the failure pattern of a cross-active-fault tunnel and propose measures for anti-dislocation design.

**Table 1.** Examples of cross-fault tunnel seismic damages.

| Number | Time | Name | Magnitude | Damage of Tunnel |
|--------|------|------|-----------|------------------|
| 1 | 1906 | San Francisco earthquake, America | 8.3 | The San Andreas dam catchment tunnel crossing the fault zone was seriously deformed by 2.4 m. Due to the influence of fault zone, the Wright No. 1 tunnel had a phenomenon of orbital uplift, and the horizontal displacement reached 1.37 m [3]. |
| 2 | 1930 | Izu earthquake, Japan | 7.3 | The sidewall of the Danah Railway Tunnel cracked seriously. Great rupture of the tunnel occurred near the fault plane. The horizontal displacement was 2.39 m and the vertical displacement was 0.6 m [7]. |
| 3 | 1971 | San Fernand earthquake, America | 6.4 | The lining of the San Fernando tunnel near the Sylmar fault was damaged and displaced on a large scale [7]. |
| 4 | 1978 | Ojima earthquake, Japan | 7.0 | The Inatori tunnel was damaged under the fault dislocation. The inverted arch and sidewall were cracked, and the concrete of the vault roof was peeled off. The steel bar was pulled off, the rail was compressed and bent, and the track and sleeper were dislocated relatively [7]. |
| 5 | 1995 | Hanshin earthquake, Japan | 7.2 | Five subway stations and approximately 3 km of subway tunnel were damaged during the earthquake. More than half of the mid-columns in the Nagata-Shinkaichi section were completely broken, resulting in roof collapse and serious land subsidence, with a maximum settlement of 2.5 m [4]. |
| 6 | 1999 | Chi-chi earthquake, Taiwan, China | 7.3 | Under the dislocation of the Chelongpu fault, the Shigangba tunnel was damaged seriously near the fault plane. The vertical deformation of the tunnel reached 4.0 m and the horizontal deformation was up to 3.0 m [5]. |

Due to the lack of actual observation data, numerical methods have been widely applied in the engineering field with the development of computational technology. In 1989, a 1D finite element model of a tunnel for soil-tunnel interaction effects was established by Burridge et al. [8]. The results showed that the mechanical response of an essentially infinite-length tunnel could be predicted. Gregor et al. [9] studied the stress response characteristics of tunnels across active faults based on finite element theory and the finite difference method. In addition, Anastasopoulos et al. [10] investigated the mechanical response of a 70 m-deep immersed tunnel crossing a normal fault in Greece, based on a nonlinear finite element model. The anti-dislocation measures of using elastic gaskets with appropriate thickness and reducing the segment length were put forward in the paper. Several studies have revealed that the continuum method is efficient and accurate, it is suitable for the study of a continuous medium and for small deformation problems. However, with these models, the failure pattern cannot be investigated from a micro perspective. More recently, researchers have been investigating the response of cross-active-fault tunnels with various discrete element models. A 3D discrete element model is used to study the dynamic response of the tunnel-fault system by Yang et al. [11], and the failure process of the tunnel is inferred according to the stress and strain distribution characteristics of the tunnel. From the relevant research, it can be found that the discrete numerical simulation demonstrates advantages in simulating large deformation problems, but with low efficiency and accuracy [12]. The discrete-continuum coupling method, which combines the advantages of the methods mentioned, is proposed. With the 3D discrete-continuum coupling approach, mechanical behaviors, such as large

deformations, can be simulated. In addition, the failure pattern can be investigated from a micro perspective, and at a greater speed [13–17].

In this paper, a 3D discrete-continuum coupling method was first introduced to study the failure mechanism of cross-active-fault tunnels. The micro parameters of the discrete particles were calibrated such that the macro mechanical behavior of the discrete particle assembly is consistent with the mechanical behavior of the in situ rock mass. On the basis of these works, the 3D coupling model was established. With the model, the longitudinal displacement profile and stress response of the tunnel lining under various fault dislocations were investigated. In addition, the failure pattern of the surrounding rock mass was studied from a micro perspective. Moreover, the design strategy for flexible joint was investigated, based on the present work. The tunnel lining structures were separated into several independent segments along the longitudinal direction, and the gaps between the lining segments were filled with materials of low strength. Then, a parametric study of the segment length, the width and the strength of the flexible joint were performed. The orthogonal array test technique was utilized to investigate the level of influence of the main design parameters of the flexible joint on the internal stress of the lining. With the obtained knowledge, the optimal combination for flexible joint design was presented.

## 2. PFC$^{3D}$/FLAC$^{3D}$ Coupling Simulation Method

The 3D discrete-continuum coupling approach is described in detail later herein.

PFC$^{3D}$ is a program based on the discrete element method with obvious advantages in simulating failure morphology of granular materials. The spherical discrete-element-method particle is a rigid body that moves independently. The solution of PFC$^{3D}$ is based on the motion equation and the contact force equation [18].

FLAC$^{3D}$ is a program based on the explicit finite difference method with advantages in computational efficiency and is useful to perform large-strain analysis for continuous media. The relationship between the force and displacement in a continuum system can be established with FLAC$^{3D}$. This allows the derivation of momentum equilibrium for each node to create equilibrium with time steps [19,20].

Taking advantage of each modeling scheme and minimizing the requirement for computational resources, a 3D discrete-continuum coupling approach was proposed. The large deformation and failure process of rock masses can be simulated from a micro scale. The coupling method involves two fundamental assumptions: (1) The macro property of the discrete particle assembly is consistent with that of the continuous model [21]. (2) The velocity and force are consistent at the contact boundary [22].

The traditional coupling between PFC and FLAC can only use a Socket I/O connection for data exchange, but there is a loss of data in this process and with low accuracy. Thus, the old version was abandoned, and the latest version of FLAC$^{3D}$ 6.0 was introduced. In FLAC$^{3D}$ 6.0, the continuous zones and discrete particles can be displayed simultaneously on stream. The wall of PFC$^{3D}$ is composed of triangular surfaces, which are made of mesh. In the triangular surfaces, the velocity and position of the triangular vertex are the functions of time. The coupling scheme logic is to transform the contact force and moment acting on the wall into the equivalent force system acting on the nodes of the triangle. Then, the force on the nodes is transferred to the gridpoint of FLAC$^{3D}$, taking into account the effect of stiffness [23,24]. The data transfer and coupling scheme of FLAC$^{3D}$/PFC$^{3D}$ are shown in Figure 1.

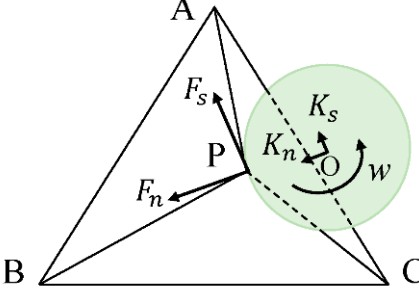

$\omega$ — Rotation angle
A — Coincident surface of PFC$^{3D}$ wall and FLAC$^{3D}$ zone
B — Spherical discrete particles
$F_n$ — Normal component of contact force
$F_s$ — Shear component of contact force
$K_n$ — Normal stiffness　　$K_s$ — Shear stiffness

(**a**)

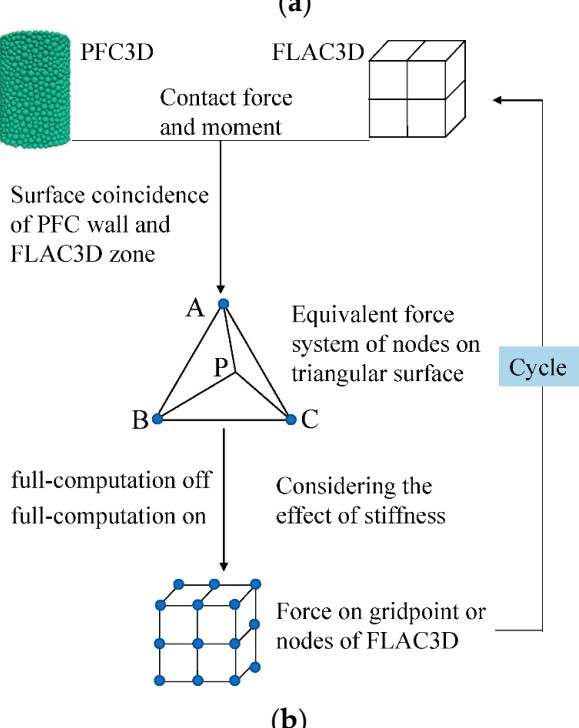

(**b**)

**Figure 1.** (**a**) Data transfer between FLAC$^{3D}$ and PFC$^{3D}$; (**b**) the FLAC$^{3D}$/PFC$^{3D}$ coupling scheme.

## 3. Numerical Model Test

### 3.1. Xianglushan Water Transmission Tunnel

The numerical model is based on a real engineering project, the Xianglushan tunnel in China. This water transmission tunnel was designed for transmitting the water from the Shigu River to the central Yunnan Province, where the water resources are short. The length of the Xianglushan tunnel is 63.426 km. Its cross section is circular, with a diameter of 8.40 m. The surrounding rock mass of the tunnel can be primarily classified into type V of the Chinese code. The site is located within the Xian Shuihe–Diandong seismic zone of the Qinghai-Tibetan region, which contains many problematic areas, including some complicated formations, areas of tectonics and severely faulted zones. There are multiple faults along the tunnel axis, and three of them are the main active faults, named Longpan—Qiaohou, Lijiang—Jianchuan and Heqing—Eryuan, as shown in Figure 2a. For the active faults, the design value of the horizontal displacement for a 100-year period is up to 2.50 m and the maximum value of the vertical displacement is 0.36 m. Among them, the Longpan-Qiaohou fault is the widest and is a normal fault (see Figure 2b). The fault is about N10° E, NW < 80°, and the intersection angle between the fault and the tunnel axis is approximately 50°, which pose a serious threat to the tunnel in the form of displacement.

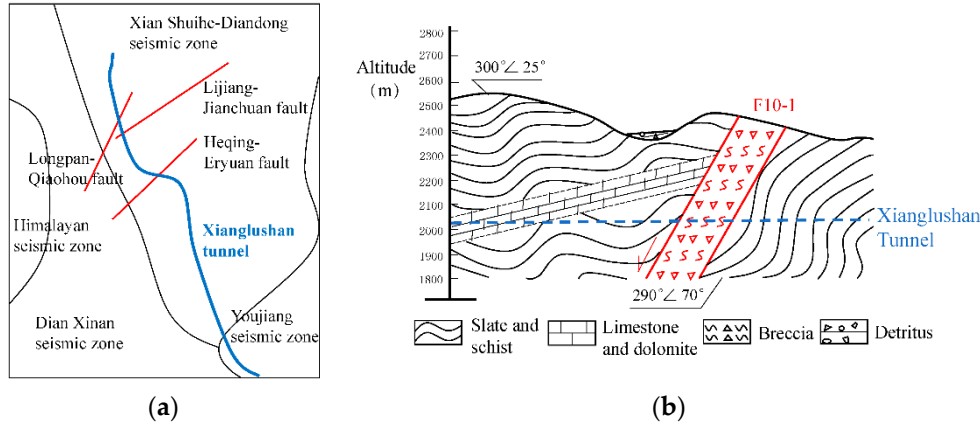

**Figure 2.** (**a**) Location of Xianglushan tunnel; (**b**) geological longitudinal profile of the Xianglushan tunnel with the Longpan-Qiaohou fault.

### 3.2. Coupling Model

In the present paper, a particular model is first presented to simulate the Xianglushan tunnel under the dislocation of the Longpan-Qiaohou fault. With it, the longitudinal displacement profile and stress response of the lining under the fault dislocations are studied. The failure pattern of the surrounding rock mass is investigated from a micro perspective. In the coupling model, the continuum method is used to simulate the tunnel and is fixed, while the discrete particle assembly is used to represent the surrounding rock mass. The intersection angle between the tunnel and the fault is approximately 50°. Meanwhile, the surrounding rock mass consists of two parts (see Figure 3): the left portion is fixed and the right portion possesses an upward velocity. Thus, the dislocation of the surrounding rock mass is simulated by the motion of the right portion.

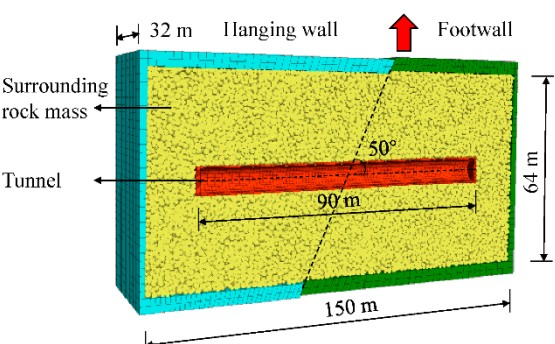

**Figure 3.** Vertical section diagram of the fault model.

Modeling of Tunnel: the FLAC Model. In the numerical model, the tunnel is simulated as the continuous zone. The tunnel section is circular. The inner and outer diameters of the section are 7 and 8 m, respectively, and are approximately equal to the diameter of the Xianglushan tunnel (see Figure 4). The lining thickness is 0.5 m. In addition, the trial calculations show that the tunnel length of 90 m is sufficient to reasonably determine the stress response of the lining. Meanwhile, the lining is assumed to be elastic, and C30 concrete is used as the material in the model, which has a Young's modulus of 30 GPa and Poisson's ratio of 0.2.

Modeling of Surrounding Rock Mass: the PFC Model. Owing to the limited calculation capacity, only approximately 150,000 discrete particles could be used to simulate the surrounding rock mass. It is generally believed that when the surrounding rock range exceeds the tunnel radius by 6 to 10 times, the boundary conditions would have little effect on the numerical calculation results. In this paper, the surrounding rock range is selected as six times the tunnel radius. Thus, the volume with the dimensions of 150 m × 64 m × 64 m outside the tunnel is filled with discrete particles. By considering

the volume outside the tunnel and the number of discrete particles, the diameter of the discrete particles is determined to be between 0.5 and 1.0 m.

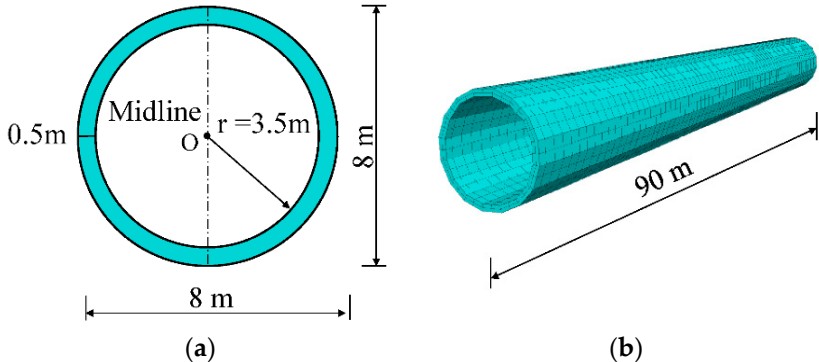

**Figure 4.** Sketch of tunnel in faulting model. (**a**) Cross section diagram; (**b**) grid division.

Then, the specific micro parameters of the particles are assigned. In the discrete particle assembly, a specified set of micro parameters corresponds to only one macro feature mode [25,26]. The macro mechanical behaviors of the in situ rock mass are obtained according to the engineering data and results from previous studies. In this model, the macro mechanical behaviors of the discrete particle assembly are guaranteed to be consistent with the mechanical behavior of the in situ rock mass by parameter calibration [27,28]. With a specified set of micro parameters of the particles, the triaxial compression tests are carried out on the model to obtain the test values under various confining pressures. From the final micro parameters of the discrete particle assembly shown in Table 2, in the triaxial compression test, the Young's modulus of the model is measured as E = 6.7 GPa, and the Poisson's ratio is measured as $\mu$ = 0.25, which are similar to the macro mechanical parameters of the surrounding rock mass, listed in Table 3. It is observed that the macro mechanical behaviors of the discrete particle assembly are the same as those of the surrounding rock mass in the Xianglushan tunnel. The model thus accurately reflects the strength and deformation behaviors of the in situ rock mass.

**Table 2.** Micro mechanical parameters of discrete particles.

| Parallel Bond Group | | | Parallel Bond Contact Model | | |
|---|---|---|---|---|---|
| Minimum particle radius (m) | Maximum particle radius (m) | Effective modulus (GPa) | Effective modulus (GPa) | Tensile strength (MPa) | Cohesion(MPa) |
| 0.50 | 1.00 | 3.00 | 3.00 | 0.60 | 1.00 |
| Ball density (kg·m$^{-3}$) | Friction coefficient | Normal-to-shear stiffness ratio | Normal-to-shear stiffness ratio | Friction angle (°) | Normal critical damping ratio |
| 2600 | 0.30 | 1.00 | 3: 5 | 0.00 | 0.20 |

**Table 3.** Macro mechanical parameters of the surrounding rock mass.

| Name | Effective Modulus (GPa) | Poisson's Ratio | Cohesion (MPa) | Friction Angle (°) | Compression Strength (MPa) |
|---|---|---|---|---|---|
| Surrounding rock mass of Xianglushan tunnel | 6.7 | 0.25 | 0.8 | 40 | 45 |

## 4. Computations and Results

In the numerical model, the deformation and stress response of the tunnel lining and the distribution of the cracks are investigated under the conditions of fault dislocation. By referring to the design strategy of the Xianglushan tunnel, the magnitudes of the fault dislocation are considered as 0.2, 0.4, 0.6, 0.8, and 1.0 m in this model. It can be seen from the following plots that the stress and maximum deformation are mainly concentrated at the bottom of the tunnel lining. Therefore, the lining bottom is selected as the subject of investigation.

### 4.1. Deformation and Stress Response of the Lining

The fault displacement of 0.6 m is taken as an example case. The continuity of the displacement between the tunnel and surrounding rock mass is observed from the displacement contour shown in Figure 5a. Meanwhile, the vertical displacement curves of the tunnel lining under different fault dislocations are shown in Figure 5b, where the horizontal axis shows the longitudinal position of the tunnel, and the vertical axis represents the vertical displacement of the tunnel. As shown in the curves, the displacement values of the tunnel in the footwall gradually increase with the increase of the magnitude of the fault dislocation, and a stable phenomenon in the hanging wall. The main deformation of the tunnel lining is concentrated within a limited range of 30 m along the tunnel axis on both sides of the fault plane, and the influence range is not affected by the magnitude of the fault dislocation.

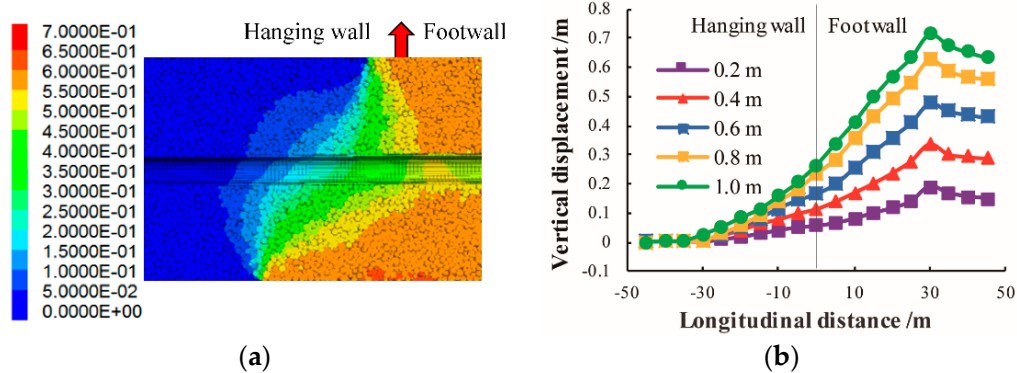

(**a**)　　　　　　　　　　　　　　　　　　　　　　　　(**b**)

**Figure 5.** (**a**) Displacement contour of the coupling model under the fault displacement of 0.6 m; (**b**) vertical displacement curves of the tunnel under different magnitudes of the fault dislocation.

The axial stress curves of the tunnel under different magnitudes of fault dislocation are shown in Figure 6a; a positive value indicates tensile stress. Figure 6b shows the shear stress curves of the tunnel. It can be seen that the lining in the hanging wall is mainly subjected to tensile stress while the lining in the footwall is primarily compressed. The stress increases with the fault dislocation. Particles near the fault plane exhibit stronger effects on the tunnel.

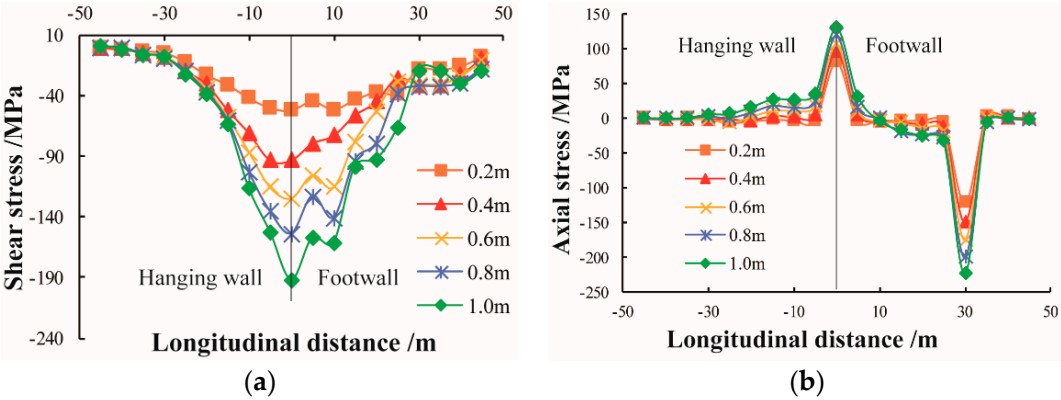

(**a**)　　　　　　　　　　　　　　　　　　　　　　　　(**b**)

**Figure 6.** Internal stress distributions along tunnel axis. (**a**) Shear force; (**b**) axial force.

### 4.2. Failure Process

The development and distribution of cracks are shown in Figure 7. With fault dislocation, the number of cracks near the fault increases rapidly. Figure 8 shows the number of various crack types under different magnitudes of the fault dislocation, which indicates that shear failure is the main failure pattern of the surrounding rock mass.

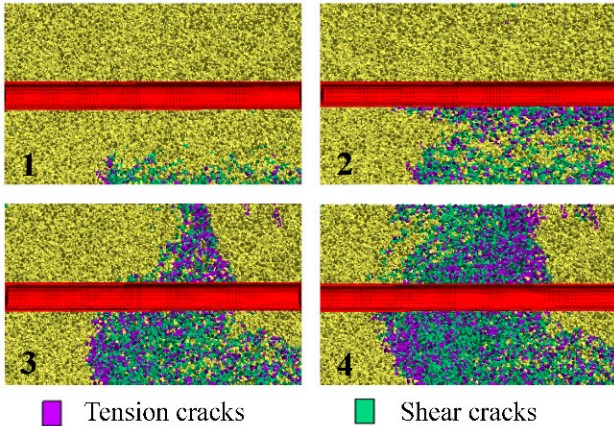

**Figure 7.** Crack development and distribution during fault displacement.

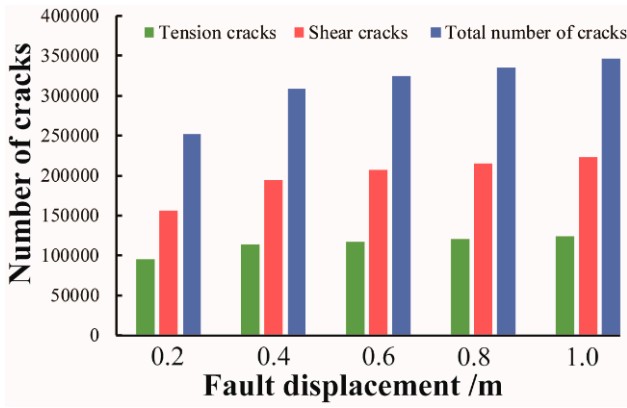

**Figure 8.** Number of various cracks at different fault distances.

## 5. Design Strategy of the Flexible Joint

### 5.1. Flexible Lining

Referring to the collected construction tunnels, some engineering cases adopt this approach [29,30]. Due to the lack of systematic design theory and calculation methods, the design parameters of flexible joint are usually determined empirically so that the application and development of flexible joint design are restricted.

In this paper, the design strategy of flexible joint for the cross-fault tunnel was investigated with the present numerical model. The tunnel lining is considered to be continuous zones and is separated into several independent sections along the longitudinal direction; materials with low strength are used to fill the gaps between the lining sections [31]. The lining structure with flexible joint is shown in Figure 9. The main parameters of the tunnel and the surrounding rock mass are shown in Table 4.

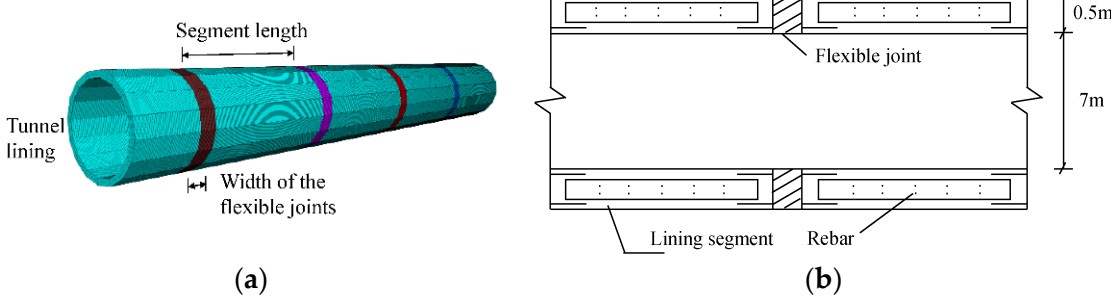

**Figure 9.** (**a**) Lining structure with flexible joints; (**b**) section profile of flexible lining of Xianglushan tunnel.

**Table 4.** Parameters of materials.

| Name | Density/(kg·m⁻³) | Young's Modulus/(GPa) | Poisson's Ratio | Internal Friction Angle/(°) | Cohesion/(MPa) |
| --- | --- | --- | --- | --- | --- |
| Surrounding rock mass | 2200 | 6.7 | 0.25 | 40 | 0.8 |
| Lining segment | 2500 | 30.0 | 0.25 | - | - |
| Flexible material | 2000 | Pending | 0.25 | - | - |

Meanwhile, in order to present the influence of design parameters of the flexible joint more clearly, and to better show the stress response and deformation characteristics of the lining under different design parameters, in this model, the fault displacement is taken as 1.0 m. The longitudinal displacement profile and stress response under the fault displacement of 1.0 m are shown in Figures 10 and 11, respectively. It is obvious that the application of the flexible joint makes the lining structure more flexible. The vertical displacement curve of the lining with flexible joints is not very smooth but slightly stepped. This is because the deformation of the material in the gaps between the lining segments is smaller than the deformation of the lining segments. Thus, the application of the flexible joint is an effective measure to accommodate the displacement of lining caused by fault dislocation. In addition, since the flexible connecting material with low strength can only transmit a small internal stress, the stress value at the flexible joint is small so that the stress loading on the lining structure is reduced [32].

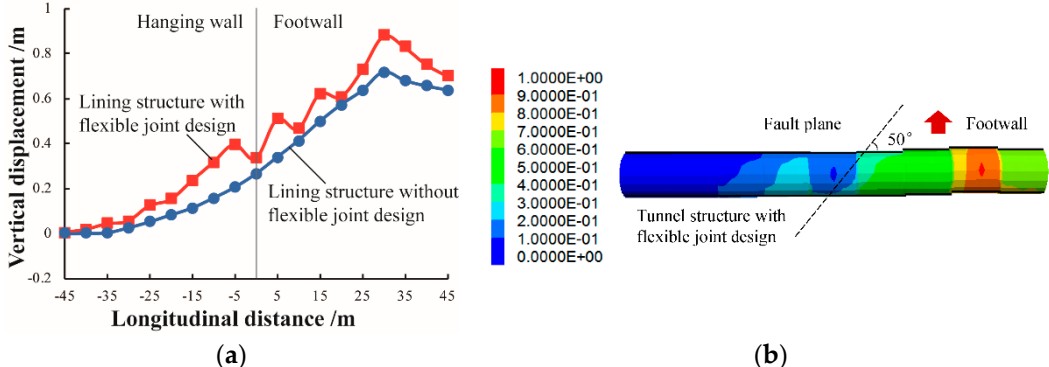

(**a**)                                        (**b**)

**Figure 10.** Displacement response (**a**) Comparison of lining with and without flexible joints; (**b**) displacement contour.

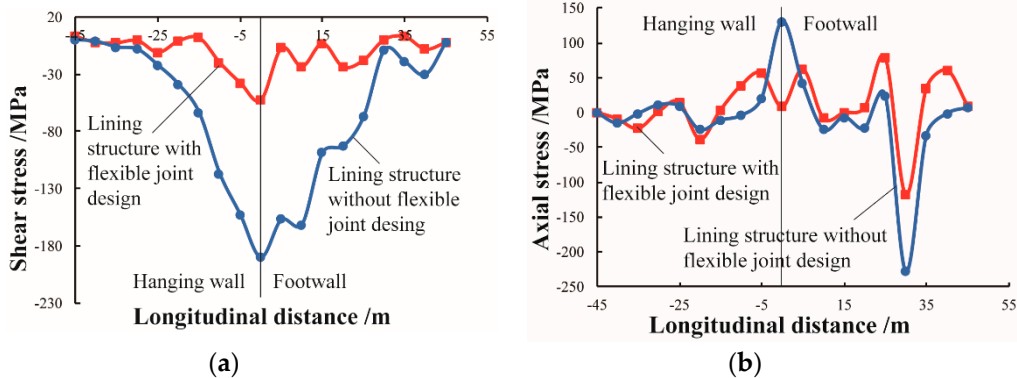

(**a**)                                        (**b**)

**Figure 11.** Stress response. (**a**) Shear stress; (**b**) axial stress.

## 5.2. Parameter Analysis

The length of the lining segments and the width and the strength of the flexible joints have a great influence on the stress and deformation characteristics of the lining. In previous studies, however, the main design parameters of flexible joint have been determined based on engineering experience,

and the calculation method is not yet mature. Therefore, it is particularly important to determine the length of the lining segment, the width and the material strength of the flexible joint.

Based on previous research results, the length of the concrete lining trolleys used in tunnel construction is generally 6–15 m. In this paper, the lengths of the lining segments are taken as 6 m, 10 m, and 15 m, respectively. To investigate the mechanism of the influence of the width of flexible joint, the widths of the flexible joint are taken as 0.5 m, 1.0 m, and 1.5 m, respectively. In addition, the material strength of the flexible joint is determined according to the ratio of the strength of the flexible joint to the strength of the lining segment, which are taken as 1/10, 1/50, and 1/100, respectively. The main design parameters and corresponding levels of flexible joints are shown in Table 5. The parameter study is carried out, keeping the magnitude of the fault dislocation at 1.0 m.

**Table 5.** The main design parameters and corresponding levels of flexible joints.

| Level | Segment Length/m | Width of Flexible Joint/m | Ratio of Strength of the Flexible Joint Material to the Lining Segment |
|-------|------------------|---------------------------|-----------------------------------------------------------------------|
| 1 | 6 | 0.5 | 1/10 |
| 2 | 10 | 1.0 | 1/50 |
| 3 | 15 | 1.5 | 1/100 |

Segment length: The deformation and stress response of the lining are studied in the cases where the width of the flexible joint is 1.0 m, the ratio of the strength of flexible joints to the strength of lining segments is 1/100, and the lengths of the lining segments are 6 m, 10 m and 15 m, respectively. Figure 12a shows the vertical displacement curve of tunnel with different segment lengths. Figure 12a,b show the stress response of the tunnel lining. It is obvious that the displacement of the lining increases with the decrease of the segment length. The lining structure will be more flexible to accommodate the displacement caused by fault dislocation with the smaller segment length. In addition, the stress of the tunnel lining is concentrated within a limited range of 30 m along the tunnel axis on both sides of the fault plane. The peak stress decreases with the decrease of the length of the lining segments. Therefore, the following conclusion is verified: The smaller segment length of the tunnel lining is beneficial to absorb the displacement generated under the fault dislocation, and the internal stress of the lining can be reduced to ensure the stability of the lining structure.

Width of flexible joint: The deformation and stress response of the lining are studied in cases where the length of the lining segment is 1.0 m, the ratio of the strength of the flexible joints to the strength of the lining segments is 1/100, and the widths of the flexible joints are 0.5 m, 1.0 m and 1.5 m, respectively. The vertical displacement curves with different widths of flexible joint are shown in Figure 13a. Figure 13b,c show the shear stress and the axial stress of the tunnel with different widths of flexible joints. It is found that the vertical displacement of the lining increases with the increase of the width of the flexible joint. At the same time, increasing the width of the flexible joint can reduce the stress loading on the lining. However, it can be seen that there is small difference between these curves, indicating that the width of the flexible joint has little influence on the stress and deformation of the lining.

Strength of flexible joint: This section studies the deformation and stress response of the lining in cases where the length of the lining segment is 1.0 m, the width of the flexible joints is 1.0 m, and the ratios of the strength of the flexible joints to the strength of the lining segments are 1/10, 1/50 and 1/100. Figure 14a shows the vertical displacement curve with different material strengths of the flexible joint. Figure 14b,c show the shear stress and the axial stress of the tunnel with different strengths of the flexible joints, respectively. From the comparison of these curves, we can see that the weaker material strength of the flexible joints is beneficial to accommodate the displacement generated under the fault dislocation, reduce the internal stress of the tunnel lining, and increase the tunnel flexibility.

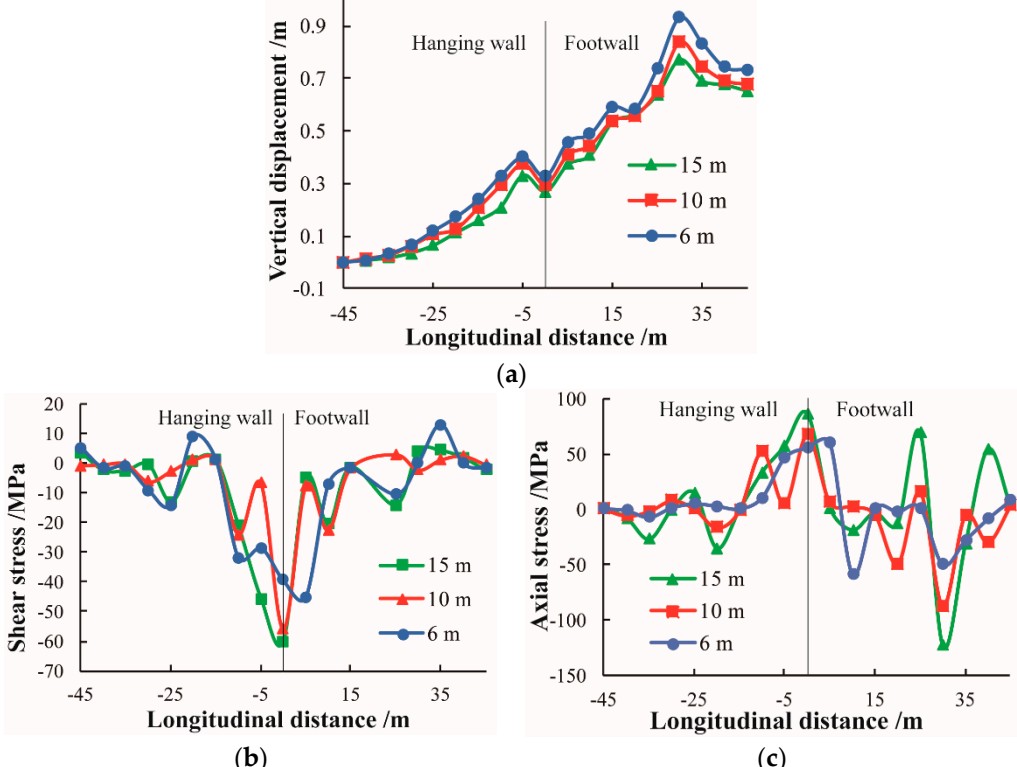

**Figure 12.** Stress and displacement curves of the lining with flexible joints of different segment length. (**a**) Vertical displacement; (**b**) shear stress; (**c**) axial stress.

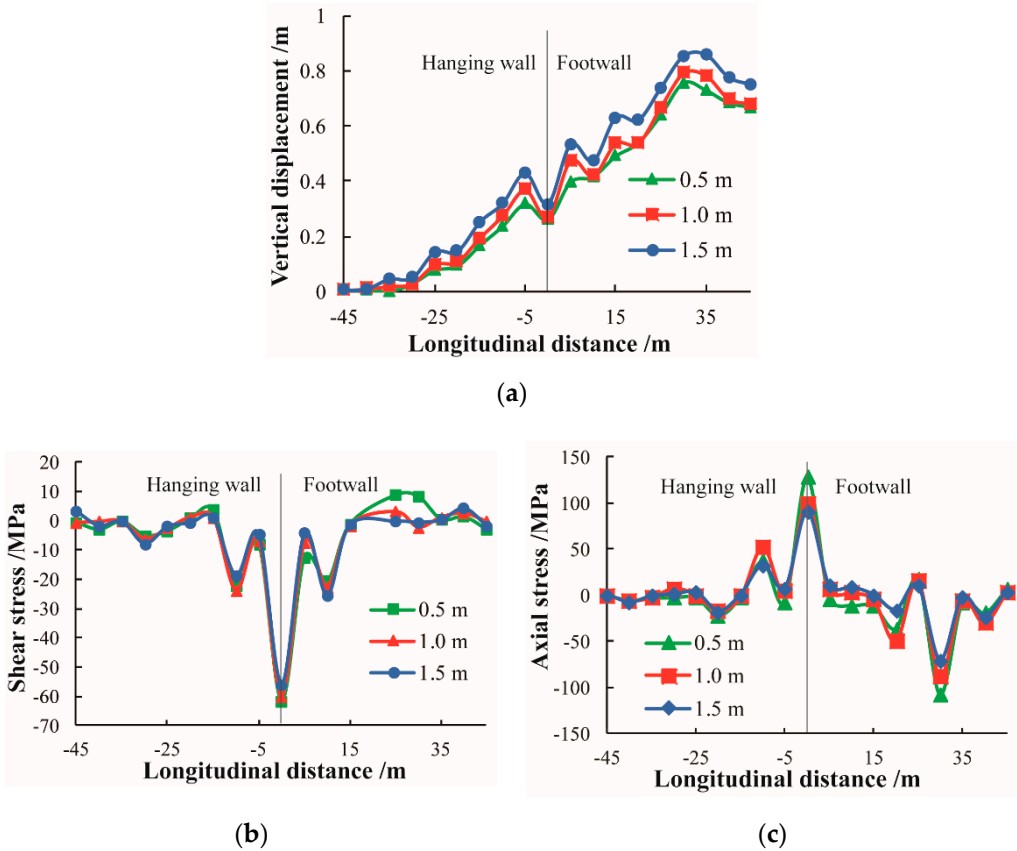

**Figure 13.** Stress and displacement curve of the lining with flexible joints of different joint widths. (**a**)Vertical displacement; (**b**) shear stress; (**c**) axial stress.

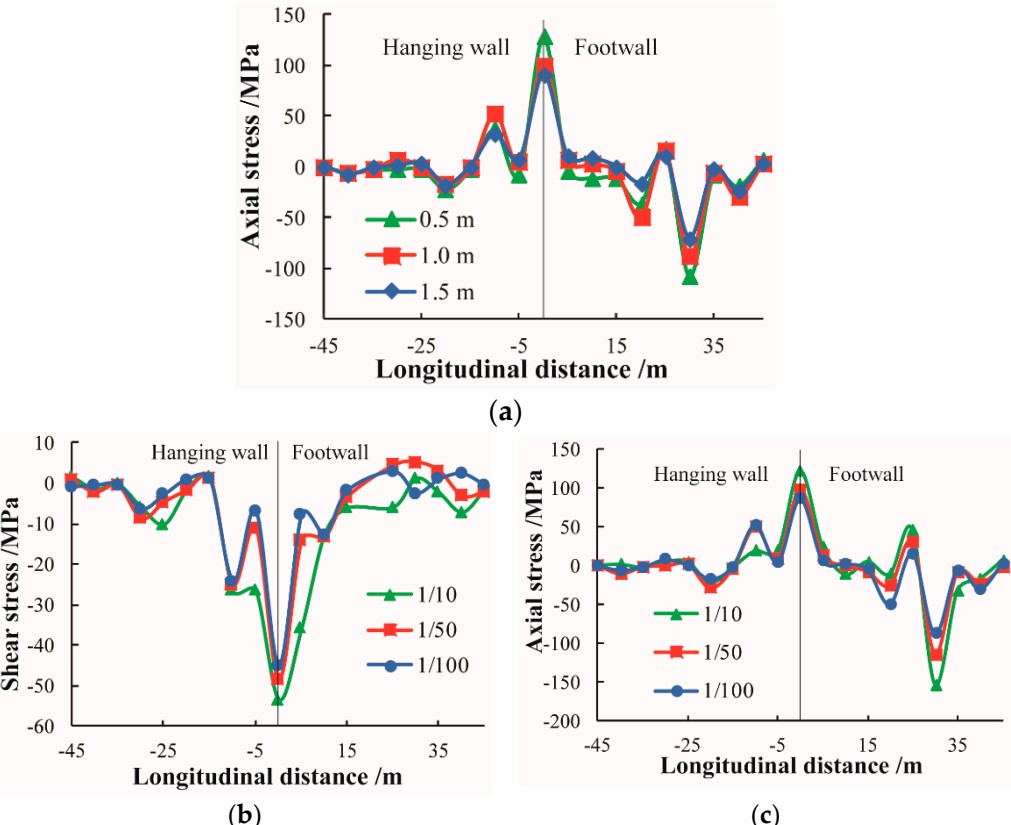

**Figure 14.** Curves of the lining with flexible joints of different joint strengths. (**a**) Vertical displacement; (**b**) shear stress; (**c**) axial stress.

*5.3. Orthogonal Array Test*

The orthogonal array test technique was utilized to investigate the magnitude of influence of the main design parameters of the flexible joint on the lining stress. Furthermore, the optimal combination for flexible joint design was presented by range analysis.

Referring to the design parameters and the corresponding levels in Table 5, the orthogonal array test was carried out with an $L9(3^4)$ orthogonal array table, and the last one was listed as a blank column [33,34]. A total of nine tests were carried out. The distribution curves of the shear stress and axial stress are shown in Figure 15, which can be compared with the stress response of the lining without flexible joints in Figure 11. It can be seen that under the fault dislocation of 1.0 m, the shear stress and axial stress distribution of the lining with flexible joints are basically the same as the distribution of the lining without flexible joints. Additionally, the application of the flexible joint can greatly reduce the internal stress of the tunnel lining.

Table 6 shows the reduction of the maximum shear stress and maximum axial stress in the nine tests. It can be seen that different combinations of the main parameters have different effects on the reduction of the internal stress. In this paper, $K_i$ ($I$ = 1, 2, 3) denotes the test index of each parameter at each level; here, the test index refers to the reduction of the shear stress and axial stress, expressed in a percentage. $\overline{K_i}$ is the average value of $K_i$, and it can judge the optimal combination of the design parameters and the corresponding levels. R stands for the difference between each level of the same parameter, which is calculated by Equation (1). It can reflect the primary and secondary relationship of the parameters.

$$R = \max \overline{K_1}, \overline{K_2}, \overline{K_3} - \min \overline{K_1}, \overline{K_2}, \overline{K_3} \tag{1}$$

Table 7 lists the sum value $K_{Ei}$, the average value $\overline{K_{Ei}}$, and the range value $R_E$ for the shear stress and the axial stress at different parameters and corresponding levels. In the table, $A_i$, $B_i$ and $C_i$

($i$ = 1, 2, 3) represent the segment length, the flexible joint width, and the material strength of flexible joint, respectively. For example, the influence of the different segment lengths on the test index is recorded by $\overline{K}_1$, $\overline{K}_2$ and $\overline{K}_3$ in line A. It can be seen from the table that $\overline{K}_1$, $\overline{K}_2$ and $\overline{K}_3$ are obviously different, which shows that the average value reflects the influence of the same parameter and different levels on the test index. Similarly, the influence of the flexible joint width and the flexible joint strength on the test index will also be manifested by $\overline{K}_1$, $\overline{K}_2$ and $\overline{K}_3$ in lines B and C. In the range analysis, the larger the test index value is, the greater the reduction of the lining stress will be, and the safer the lining structure will be.

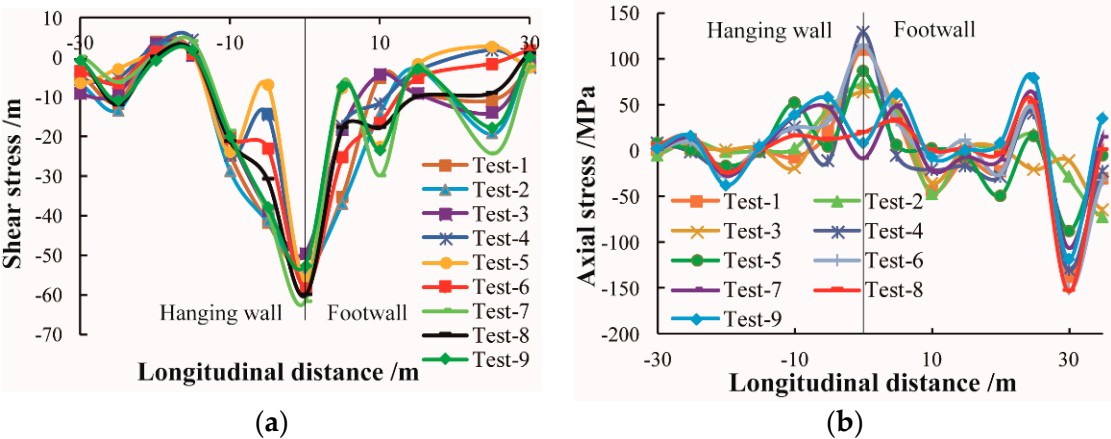

(**a**)　　　　　　　　　　　　　　　　　　　　　(**b**)

**Figure 15.** Internal stress distribution along tunnel axis in orthogonal array test. (**a**) Shear stress; (**b**) axial stress.

**Table 6.** Results of the orthogonal array test.

| Number | Parameters and the Corresponding Levels | | | Results | |
|---|---|---|---|---|---|
| | Segment Length/m | Width of the Flexible Joint/m | Ratio of the Strength of Flexible Joint to the Strength of Lining Segments | Reduction of Maximum Shear Stress/% | Reduction of Maximum Axial Stress/% |
| 1 | 6 | 0.5 | 1/10 | 71.3 | 39.5 |
| 2 | 6 | 1.0 | 1/50 | 71.9 | 68.9 |
| 3 | 6 | 1.5 | 1/100 | 72.8 | 72.0 |
| 4 | 10 | 0.5 | 1/50 | 69.5 | 43.4 |
| 5 | 10 | 1.0 | 1/100 | 70.5 | 61.9 |
| 6 | 10 | 1.5 | 1/10 | 69.4 | 34.2 |
| 7 | 15 | 0.5 | 1/100 | 68.4 | 53.5 |
| 8 | 15 | 1.0 | 1/10 | 68.3 | 32.9 |
| 9 | 15 | 1.5 | 1/50 | 72.2 | 48.7 |

**Table 7.** Values of $K_i$, $\overline{K}$ and $R$ at different design parameters.

| | Values Concerning the Shear Stress | | | | | | | Values Concerning the Axial Stress | | | | | | |
|---|---|---|---|---|---|---|---|---|---|---|---|---|---|---|
| | $K_{E1}$ | $\overline{K}_{E3}$ | $K_{E3}$ | $\overline{K}_{E1}$ | $\overline{K}_{E2}$ | $\overline{K}_{E3}$ | $R_E$ | $K_{F1}$ | $K_{F2}$ | $K_{F3}$ | $\overline{K}_{F1}$ | $\overline{K}_{F2}$ | $\overline{K}_{F3}$ | $R_F$ |
| A | 216.0 | 209.4 | 208.9 | 72.0 | 69.8 | 69.6 | 2.4 | 180.4 | 139.5 | 135.1 | 60.1 | 46.5 | 45.0 | 15.1 |
| B | 209.2 | 210.7 | 214.4 | 69.7 | 70.2 | 71.5 | 1.8 | 136.4 | 163.7 | 154.9 | 45.4 | 54.6 | 51.6 | 9.2 |
| C | 209.0 | 210.6 | 214.7 | 69.7 | 70.2 | 71.6 | 1.9 | 106.6 | 161.0 | 187.4 | 35.5 | 53.7 | 62.5 | 27.0 |
| | Primary and secondary order: A > C > B | | | | | | | Primary and secondary order: C > A > B | | | | | | |

The tendency chart, which reflects the influence level of each parameter, is shown in Figure 16 based on Table 7. With it, the optimal combination of the design parameters for the flexible joint is

obtained: The optimal combination concerning shear stress reduction is $A_1B_3C_3$, and the optimal combination for axial stress reduction is $A_1B_2C_3$.

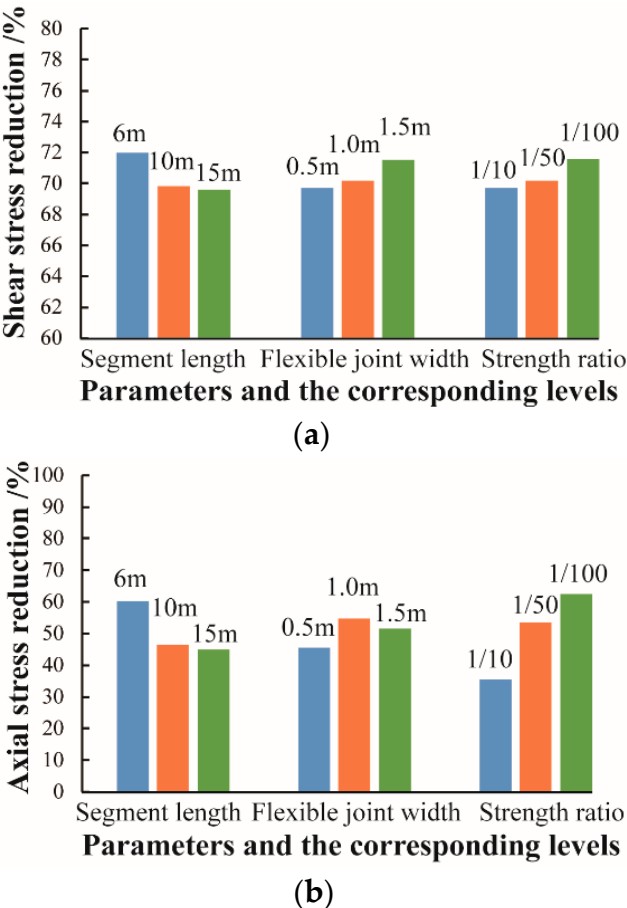

**Figure 16.** Tendency chart of the design parameters. (**a**) Shear stress reduction; (**b**) axial stress reduction.

Since the two optimal combinations are not consistent, the optimal level should be determined according to the primary and secondary relationship of the parameters. In this test, level $B_3$ is the optimal level according to the influence of the shear reduction. While considering the axial stress reduction, the $B_2$ level is the best choice. However, referring to the above results, the width of the flexible joint has less influence on the test index. When the width of the flexible joint is taken as $B_2$, the reduction of the shear stress is 1.82%, which is lower than that of $B_3$, and the reduction of the axial stress is 5.81%, which is higher than that of $B_3$. Moreover, gaps between the lining segments that are too wide are not conducive to the safety of the lining structure.

Therefore, considering the current lining molding technology, the optimal combination of flexible joint is $A_1B_2C_3$. It is suggested that a reasonable segment length is 6 m, a reasonable width of the flexible joint is 1.0 m, and a reasonable material strength of the flexible joint is 10% of the strength of the lining segments.

## 6. Conclusions

This study investigates the rupture problem of a cross-active-fault tunnel with a 3D discrete-continuum coupling approach. On the basis of these analyses, the following relevant conclusions are drawn:

(1)  In this coupling model, the macro mechanical behaviors of the discrete particle assembly are calibrated to be consistent with the mechanical behavior of the in situ rock mass. In addition, the validity of the coupling model is demonstrated by the observed continuity on the displacement contour of the surrounding mass and the tunnel.

(2) With fault dislocation, the displacement and internal stress of the tunnel lining increase correspondingly. The main deformation and stress response of the tunnel lining is concentrated within a limited range of 30 m along the tunnel axis on both sides of the fault plane. This influence range is not affected by the magnitude of the fault dislocation.

(3) The failure pattern of the surrounding rock mass was investigated from a micro perspective. The shear failure is the primary failure pattern of the surrounding rock mass. With fault dislocation, the number of cracks dramatically increases.

(4) Under fault displacement, the tunnel lining with flexible joints will be more flexible to accommodate the displacement and reduce the stress imposed on the lining segments. In addition, a smaller segment length, wider width and weaker strength of the flexible joint are beneficial to the tunnel anti-dislocation performance.

(5) For the Xianglushan tunnel under a fault dislocation of 1.0 m, it was determined from the orthogonal array test and the range analysis that a reasonable segment length is 6 m, a reasonable width of the flexible joint is 1.0 m, and a reasonable material strength of the flexible joint is 10% of the strength of the lining segments.

(6) These findings may provide some reference for the anti-dislocation design of a cross-active-fault tunnel. However, it is recognized that the work described in this paper is based on several approximations. For example, only the elastic model of the tunnel is considered for the sake of simplification; therefore, the issue will be studied in detail in future studies.

**Author Contributions:** The paper was written by Y.M. under the guidance of Q.S. and C.Z. The formal analysis and pre-literature research were carried out by Y.M. and Q.S. The coupling simulation method was proposed and by Y.M. and Z.C. The numerical model test was carried out by Y.M. under the help of G.Z.

**Funding:** The study was financially supported by the National Key R&D Program of China (No. 2016YFC0401803), the Key Laboratory for GeoMechanics and Deep Underground Engineering, China University of Mining & Technology (No. SKLGDUEK1912), the National Basic Research Program of China (No. 2015CB057905), the National Natural Science Foundation of China (Nos. 51779253, 41672319), and the Provincial Natural Science Foundation of Hubei (No. 2017CFB725).

**Conflicts of Interest:** The authors declare that they have no conflict of interest.

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
