# Peer review of "A 3D Discrete-Continuum Coupling Approach for Investigating the Deformation and Failure Mechanism of Tunnels across an Active Fault: A Case Study of Xianglushan Tunnel"

_applsci, doi:10.3390/app9112318_

Reviewer 1 Report

This study investigates the rupture problem of a cross-active-fault tunnel with a 3d discrete-continuum coupling approach. In this paper, the longitudinal displacement profile and stress response of the tunnel lining were studied under various fault dislocations. The failure pattern of the surrounding rock mass was investigated from a micro perspective. Meanwhile, the design strategy of the flexible joint was investigated with the present numerical model. And the optimal combination for the flexible joint design was presented. Findings may provide references for the anti-dislocation issue of tunnels across active faults. I believe this paper can be published after a few corrections.

Page 2, line 80 and line 87: It is suggested to simplify section 2.1 and section 2.2, and merge them into the part of PFC3D/FLAC3D Coupling Simulation Method.

Table. 2: How did the author get the micromechanical parameters of discrete particles in Fig.2? And brief illustrations or explanations about parameters in Table. 2 would be better than nothing afterwards.

Page 15, lines 387: The last sentence, that a reasonable material strength of the flexible joint is 1% of the strength of the lining segments, is not referred to in the paper. Also, I believe that the 1% should be 10%.

Page 15, lines 411: The same error as the last comment. Please check the conclusion, which is one of the most important parts of the whole paper.

Author Response

Dear Reviewer,

Thank you for your comments concerning our manuscript entitled “A 3d Discrete-Continuum Coupling Approach for Investigating the Deformation and Failure Mechanism of Tunnels Across an Active Fault: A Case Study of Xianglushan Tunnel” (applsci-511187). Those comments are all valuable and very helpful for revising and improving our paper, as well as the important guiding significance to our researches. We have studied the comments carefully and have made correction which we hope meet with approval. The main corrections in the paper and the responds to your comments are as following:

1. Page 2, line 80 and line 87: It is suggested to simplify section 2.1 and section 2.2, and merge them into the part of PFC3D/FLAC3D Coupling Simulation Method.

Response: Thank you for your advice. These parts have been corrected in the revised paper according to your suggestion. We have simplified these two parts, and merge them into the part of PFC3D/FLAC3D Coupling Simulation Method.

2. Table. 2: How did the author get the micromechanical parameters of discrete particles in Fig.2? And brief illustrations or explanations about parameters in Table. 2 would be better than nothing afterwards.

Response: We are sorry to confuse you. The purpose of this part is to assign the specific micro parameters of the particles. It is generally known that in the discrete particle assembly, a specified set of micro parameters corresponds to only one macro feature mode. So, with the micro parameters of the discrete particle assembly shown in Table 2, the triaxial compression tests are carried out on the model to obtain the test values under various confining pressures. In this way, the macro mechanical parameters of the surrounding rock mass can be obtained, which are shown in Table 3. It is observed that the macro mechanical behaviors of the discrete particle assembly are the same as those of the surrounding rock mass in the Xianglushan tunnel. Perhaps our expression is not clear enough before. We have made minor adjustments and modifications in the article. Please have a look.

3. Page 15, lines 387: The last sentence, that a reasonable material strength of the flexible joint is 1% of the strength of the lining segments, is not referred to in the paper. Also, I believe that the 1% should be 10%.

Response: We are sorry to make such a mistake. Through the numerical model test, the reasonable material strength of the flexible joint is 10% of the strength of the lining segments. 1% has been replaced by 10% in this paper. Thank you for your careful review.

4. Page 15, lines 411: The same error as the last comment. Please check the conclusion, which is one of the most important parts of the whole paper.

Response: Once again, we are sorry to make such a mistake. We have made amendments in the article. In the future, we will treat articles with a serious attitude to avoid such mistakes.

We tried our best to improve the manuscript and made some changes in the revised manuscript according to your comments. We appreciate for your warm work earnestly, and hope that the correction will meet with approval. Once again, thank you very much for your comments and suggestions.

Best regards,

Yours sincerely,

Yalina Ma

Reviewer 2 Report

Reference: applsci-511187

Title: A 3d Discrete-Continuum Coupling Approach for Investigating the Deformation and Failure Mechanism of Tunnels Across an Active Fault: A Case Study of Xianglushan Tunnel

In this manuscript, the authors presented DEM-FDM coupling approach to study the Xianglushan Tunnel case study.

Following comments associated with the paper are presented:

·         It is important to establish the importance of the DEM-FDM coupled approach more clearly.

·         In literature, there are many DEM-FDM/FEM couple approach. What is the novelty of the present approach compared to others? Or, authors need to mention, they did not develop/propose the coupling method, but used someone else approach.

·         Suddenly, the orthogonal array test came into the picture. What is the importance of this method and how it is relevant to the coupling approach?

·         In Table 1, Reference is missing. It is important to include references.

·         Line # 54-55 are not clear. What does continuum method mean? FDM/FEM, if yes, then how the method is free from boundary conditions? How it is relevant to geometry, whether complex or simple? Boundary conditions should be Neumann boundary condition or Dirichlet boundary condition. Additionally, what is the significance to study the micro mechanics of the failure conditions?

·         Line # 61-63 are also not clear. Most cases, the DEM approach is qualitative and for quantitative approach, someone needs rigorous parametric calibration, which is completely absent in this manuscript. The DEM approach cannot supersede the FDM/FEM. Because the later one is in quantitative level.

·         Line # 88-89 are confusing. The Hooke’s law is for elastic state, while the authors mentioned they used DEM to capture failure state micro-mechanics. It is important to make sure, whether the authors used PFC3D’s built-in function or they deduced something new through FISH or user interface new model.

·         Line # 105 is not clear. Why there is a loss of data? Is there any accuracy issue?

·         Line # 108, the triangular surface made of what? Mesh or particles?

·         Figure 1 needs more attentions. It does not represent Table 2.

·         In page 5, how the fault is modeled? 1D line element or 2D? in Line # 145, what is the source of the velocity in practical field?

·         Line # 153-154, are not clear. Does it mean to avoid the boundary condition effect?

·         Mechanism to generate particles is absent. random generation or particle size distribution? In tunnel materials PSD has significant importance.  

·         Line # 171, compression test. Is it drained or undrained? It looks like tunnel’s fluid part is ignored, which is the most important part for the failure.

·         Finally, in DEM and FDM sections, the governing equations are absent. It needs to be incorporated.

This manuscript requires major revision.

Author Response

Dear Reviewer,

We appreciate for your warm work earnestly. Those comments are all valuable and very helpful for revising and improving our paper, as well as the important guiding significance to our researches. We have studied the comments carefully and have made correction which we hope meet with approval.

Please review the manuscript uploaded in the form of the attachment.

Thank you and best regards.

Yours sincerely,

Yalina Ma

Reviewer 3 Report

Really good paper.

Author Response

Dear Reviewer,

Thank you for your recognition of our manuscript entitled “A 3d Discrete-Continuum Coupling Approach for Investigating the Deformation and Failure Mechanism of Tunnels Across an Active Fault: A Case Study of Xianglushan Tunnel” (applsci-511187).

We have improved the manuscript and made some changes in the revised manuscript. And these changes will not influence the content and framework of the paper. We appreciate for your warm work earnestly. Once again, thank you very much for your recognition of this paper.

Best regards,

Yours sincerely,

Yalina Ma

Round  2

Reviewer 2 Report

Accepted in the present form.